



# Methodology and uncertainty estimation for measurements of methane leakage in a manufactured house

Anna Karion[1], Michael F. Link[2], Rileigh Robertson[2*], Tyler Boyle[1], Dustin Poppendieck[2]

[1]Special Programs Office, National Institute of Standards and Technology, Gaithersburg, Maryland, 20899, United States
[2]Engineering Laboratory, National Institute of Standards and Technology, Gaithersburg, Maryland, 20899, United States

*Currently at College of Engineering and Applied Science, University of Colorado, Boulder, Colorado, 80302, United States

*Correspondence to*: Anna Karion (Anna.Karion@nist.gov)

**Abstract.** Methane emissions from natural gas appliances and infrastructure within buildings have historically not been
captured in greenhouse gas inventories, leading to under-estimates, especially in urban areas. Recent measurements of these post-meter emissions have indicated non-negligible emissions within residences, with impacts on both indoor air quality and climate. As a result, methane losses from residential buildings have recently been included in the latest U.S. national inventory, with emission factors determined from a single study of homes in California. To facilitate future additional studies investigating building methane emissions, we conducted a controlled experiment to document a methodology for such measurements and
estimated associated uncertainties. We determined whole-house methane emission rates with a mass balance approach using near-simultaneous measurements of indoor and outdoor methane mole fractions at a manufactured house. We quantified the uncertainty in whole-house methane emission rates by varying the forced outdoor air ventilation rate of the manufactured house, measuring the outdoor air change rate using both sulfur hexafluoride and carbon dioxide tracers, and performing methane injections at prescribed rates. We found that the whole-house quiescent methane emission rate in the manufactured
house averaged 0.34 g d$^{-1}$ with methodological errors on the calculated emission rates to be approximately 20 % (root-mean-square-deviation). We also measured the quiescent leakage from the manufactured house over three months to find 24 % (1-sigma) variability in emissions over two seasons. Our findings can be used to inform plans for future studies quantifying indoor methane losses after residential meters using similar methods. Such quantification studies are sorely needed to better understand building methane emissions and their drivers to inform inventories and plan mitigation strategies.

## 1 Introduction

Methane is the second largest contributor to climate warming after carbon dioxide ($CO_2$), with a global warming potential (ability to trap heat in the atmosphere) 27 to 30 times greater than that of $CO_2$ on a 100-year basis (IPCC, 2021). Methane also plays a role as a precursor to tropospheric ozone (Mar et al., 2022). Nearly half of all residences in the United States use natural gas, whose chief constituent is methane, for one or more appliances within the dwelling (AGA, 2021; EIA, 2023). Leakage of
natural gas from appliances and pipes within residential dwellings, in addition to being a concern for indoor air quality, leads



to emissions that exfiltrate to the outdoor air. Assessing future global warming impacts of methane requires accurate accounting for fugitive gas leaks and losses in distribution systems. Historically, there have been two methods for calculating these leakages on a regional scale: top down (e.g., using atmospheric methane concentrations to estimate regional emissions) and bottom up (e.g., multiplying the average leakage from a pipe section by the total length of pipes in a region). Recent top-down

estimates of methane emissions from urban areas are larger than accounted for using bottom-up methods (Plant et al., 2019; Yadav et al., 2023; Lopez-Coto et al., 2020; Sargent et al., 2021). Historically, bottom-up methane emission inventories have not accounted for natural gas losses downstream of residential gas meters, possibly contributing to this discrepancy between top-down and bottom-up estimates in urban areas. Recent studies both at the appliance and whole-house level have shown that quiescent leakage of gas during periods with appliances off, emissions during on/off cycling of appliances, and emissions

during appliance use (including from potential inefficient combustion), can account for a significant portion of a region's total methane emissions (Lebel et al., 2022; Merrin and Francisco, 2019; Saint-Vincent and Pekney, 2020). Measurements of emissions from 75 homes in California indicated that typical homes have quiescent methane emission rates of <1 g d$^{-1}$, but with some as high as 10 g d$^{-1}$, with total emissions from residential natural gas accounting for approximately 15 % of California's natural gas emissions (Fischer et al., 2018). Consequently, the U.S. Environmental Protection Agency (EPA)

recently included quiescent residential emissions of methane in its 2022 inventory, using values based on the Fischer et al. study (EPA, 2022).

        Historically most indoor emission rate studies have been targeted at quantifying emissions of formaldehyde and subsets of volatile organic compounds (VOCs). The most common approach for determining emission rates for various chemicals indoors is measuring the emissions of individual sources using environmental chambers, and then using models to

extrapolate to the individual source emission rates in an entire building (Kelly et al., 1999; Liang et al., 2012; Mølhave et al., 1996). One drawback of this approach is the difficulty in ensuring the models account for all potential emission sources. Additionally, measuring emissions in environmental chambers can be labor intensive, time consuming, and expensive. Another approach to quantifying whole-house emission rates for various chemicals is the mass balance method (Hodgson et al., 2002; Offermann and Hodgson, 2011; Liang and Yang, 2013; Li et al., 2019). For methane this approach can quantify the sum of all

leaks in the entire volume of the house. The methane emission rate is determined by comparing the concentration of the chemical indoors and outdoors while knowing the outdoor air change rate (ACR) of the house. To the best of our knowledge, Fischer et al. (2018) is the only study to have quantified whole-house methane emissions, using blower door and tracer decay tests to determine the ACR in a mass balance approach. More recently, Nicholas et al. (2023) measured indoor methane leaks using a closed chamber method, demonstrating the method's accuracy in a study of 20 buildings in Greater Boston,

Massachusetts, USA. In this method, the basement of the building was sealed off and a mass balance equation applied assuming a zero ACR (i.e. that no methane exited the closed space).

        A common method for calculating ACR in a residence is tracer decay analysis using either sulfur hexafluoride (SF$_6$) or CO$_2$ as the trace gas (ASTM, 2023). Each tracer has its advantages and disadvantages. CO$_2$ has a measurable, varying outdoor background concentration while outdoor SF$_6$ concentrations are sufficiently small relative to the high concentrations



injected such that they can be neglected for the ACR determination. $SF_6$ analysis requires laboratory-grade analytical equipment, while commercial and less-expensive $CO_2$ sensors are readily available. Due to instrument detection limits and background $CO_2$ concentrations the number of moles of $CO_2$ required for a tracer test is over three orders of magnitude higher than for $SF_6$; however, the global warming potential of $SF_6$ is over 24,000 times greater than that of $CO_2$ (IPCC, 2021). Under typical instrument detection limits $CO_2$ as a tracer has a smaller global warming impact than $SF_6$. A few studies have quantified

the uncertainty in tracer-decay ACR measurements using each of these tracers. For example, Poppendieck et al. (2015) injected $SF_6$ as a tracer and estimated a 10 % measurement error in the ACR, while Huangfu et al. (2020) used $CO_2$ injections to calculate ACRs with an estimated uncertainty of 20 %.

        Here we demonstrate a mass balance method for estimating whole-house methane emission rates using a tracer decay ACR measurement and determine the uncertainty associated with the method. The 3.5-month study was performed in the

National Institute of Standards and Technology (NIST) manufactured house (Nabinger and Persily, 2008; Nabinger et al., 2010). First, we compared the whole-house ACR estimated using either $CO_2$ or $SF_6$ as a tracer while varying the operation of the forced outdoor air ventilation system of the manufactured house (i.e., exhaust and whole-house fans). Then, we injected methane at specified known flow rates and quantified the error in the mass balance calculation of the overall house methane emission rate. Finally, we calculated quiescent whole-house methane emission rates (i.e., with no injections or operation of

appliances) daily over several months, analyzing their variability to determine how representative one measurement is of long-term emissions. Our experimental goal was to develop an easily deployable and scalable means of measuring building emission rates, while documenting possible sources of error and associated uncertainties.

## 2 Methods

We measured whole-house emission rates of methane in a manufactured house (also called a mobile home in the United States)

on the NIST campus in Gaithersburg, Maryland between October 2023 and January 2024. The one-story house has three bedrooms and two bathrooms, a floor area of 133 $m^2$ and volume of 324 $m^3$. The house is equipped with a gas stove and exhaust fan in the kitchen, a gas furnace, and a whole-house exhaust fan (Fig. 1). The exhaust fans allowed control of the ACR in the house between 0.2 $h^{-1}$ and 2.5 $h^{-1}$. A single heating, ventilation, and air conditioning (HVAC) return was located at the HVAC unit, with supply air distributed throughout the house. The HVAC fan was operating throughout the experiment, to

ensure well-mixed conditions within the house. Instrumentation measured indoor and outdoor methane, $CO_2$, and indoor $SF_6$ (outdoor $SF_6$ concentrations were indistinguishable from background). These measurements allowed for the determination of the ACR in the house as well as the indoor emissions of methane.



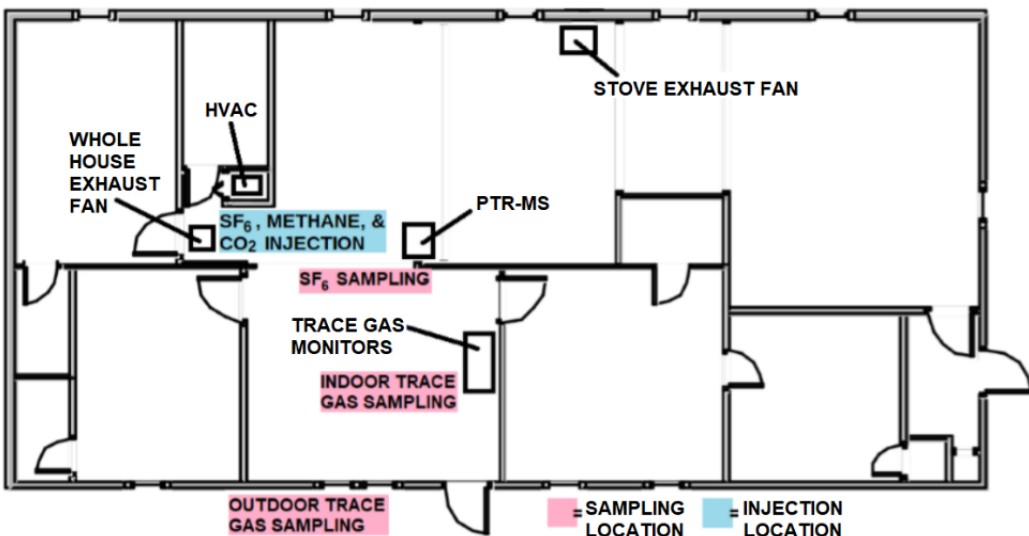

**Figure 1. Schematic of the manufactured house. Locations of injection lines (blue), gas measurement sampling (pink), and ventilation equipment are marked.**

## 2.1 Instrumentation for trace gas sampling

Mole fractions of $CO_2$ and methane were measured at approximately 2.5 s intervals by a cavity ring-down spectrometric (CRDS) analyzer and are reported here in $\mu$mol mol$^{-1}$, often also referred to as parts per million (ppm). A valve automatically switched between two different perfluoroalkoxy (PFA) 0.95 cm (3/8 inch) outer diameter inlet lines at 5-min intervals. One inlet drew outdoor air from a 1.5 m mast located 2.4 m outside the house, and the other drew air from inside the house, at locations indicated in Fig. 1. Both lines were connected to particulate filters (polytetrafluoroethylene (PTFE), 47 mm diameter, 0.45 $\mu$m pore size) at their inlets. A flush pump pulled air at 10 L min$^{-1}$ through whichever inlet line the valve was set to. Several additional instruments were simultaneously drawing air from the common side of the valve upstream of the flush pump, including the CRDS analyzer, which was set to a flow rate of approximately 70 mL min$^{-1}$. $SF_6$ was measured inside the house using a time-of-flight proton-transfer mass spectrometer (PTR-MS). $SF_6$ was quantified from the $SF_3O^+$ ion signal. The PTR-MS sampled $SF_6$ at a flowrate of 15 L min$^{-1}$ through 8 mm (inner diameter) PFA tubing with an inlet flowrate of 120 mL min$^{-1}$.

Indoor and outdoor temperature and heating and cooling activity in the house were logged by a smart thermostat. Indoor air temperatures were used to calculate air number densities. Differential pressure measurements indicated that the difference between indoor and outdoor pressures never exceeded 10 Pa, so the outdoor pressure from a weather station in Arlington, Virginia (Karion et al., 2020) was used for both indoor and outdoor molar calculations. While the HVAC fan was on for all experiments, the gas heating component of the furnace was turned off for the tracer experiments. The HVAC gas heating element was turned on in late December, cycling on and off to maintain a minimum temperature setpoint during some



of the quiescent emissions measurements. The duration that the heating element of the furnace was running was logged by the thermostat. The electrically powered whole-house air conditioning system was active during most of the experiments when the heating element was not used.

### 2.2 Injections of methane, $CO_2$ and $SF_6$

Methane, $CO_2$, and $SF_6$ were injected from standard cylinders using mass flow controllers, controlled by a custom automated

system, into the house during the measurement campaign. For the ACR calculations, a mixture of 5 % $SF_6$ (volume fraction) in nitrogen was injected at rate of 90 mL min$^{-1}$ for 2 min, achieving mole fractions of approximately 30 nmol mol$^{-1}$. Pure $CO_2$ (99.9% mole fraction) was injected at approximately 5 L min$^{-1}$ for 1 h, typically achieving mole fractions between 100 μmol mol$^{-1}$ and 600 μmol mol$^{-1}$ above outdoor values. When both $CO_2$ and $SF_6$ were injected on the same day, injections were set up to end at the same time to achieve the best comparison between ACR calculated using the two tracers. During the methane

tracer experiments, methane was injected at a constant rate for a 24-h period. The volumetric flowrate of methane from the mass flow controller was calibrated using a volumetric flow meter (manufacturer-accredited by ISO and NIST) over a range to include the methane flowrates used for the injection experiments.

### 2.3 Processing of raw mole fraction data

$CO_2$ and methane mole fraction measurements from the CRDS analyzer were initially processed as described below. The

analyzer recorded data at a resolution of 2.5 s. The values were separated into indoor and outdoor time series using the timing of the valve switching (every 5 min), which was logged separately. The first 60 s after and last 30 s prior to each valve switching event were removed to account for analyzer response time and any discrepancies in timing. For the indoor air data, the remaining measurements for each sample period were averaged. The outdoor methane measurements exhibited some high-variability time periods during which plumes of methane travelled past the inlet line for periods shorter than 5 min. Thus, for

each 5-min outdoor sampling period, the median of mole fractions was calculated instead of the mean value, in order give a more representative background around the house and reduce the influence of large, short-lived plumes. The indoor 5-min means and outdoor 5-min medians were then interpolated in time so that for each 5-min interval there was a corresponding indoor and outdoor value (i.e., using the mean of the adjacent measurements to fill the missing intervals for each). The outdoor measurements were then further smoothed using a 1-h rolling average to eliminate additional variability.

The water vapor levels inside and outside the house were different, especially during sampling in December and January. Hence, the reported dry air mole fractions needed to be converted to wet mole fractions for mass balance calculations. The internal CRDS water vapor corrected value for $CO_2$ and methane mole fractions includes a correction for both the dilution and spectroscopic influences on the measurement (Rella et al., 2013), thus we used the reported dry water-corrected values and then calculated the dilution only to obtain the true wet air mole fractions following Eq. (1):




$$X_{C,wet} = X_{C,dry}(1 - X_{H2O}), \qquad (1)$$

where $X_{H2O}$ is the mole fraction of water vapor as measured by the CRDS instrument, $X_{C,dry}$ is the dry air mole fraction of either methane or $CO_2$, and $X_{C,wet}$ is the dilution-corrected wet-air mole fraction ($X_{C,wet}$ is simplified to $X$ in subsequent

equations). The initial data processing described above resulted in a time series at 5-min intervals of indoor, outdoor, and smoothed outdoor values that were used in subsequent analyses of methane and $CO_2$. Figure 2 shows an example of the raw time series data along with the resulting processed data from Oct 8 to Oct 11, 2023, a period with three different methane injection rates along with daily $CO_2$ injection spikes for the ACR determination.

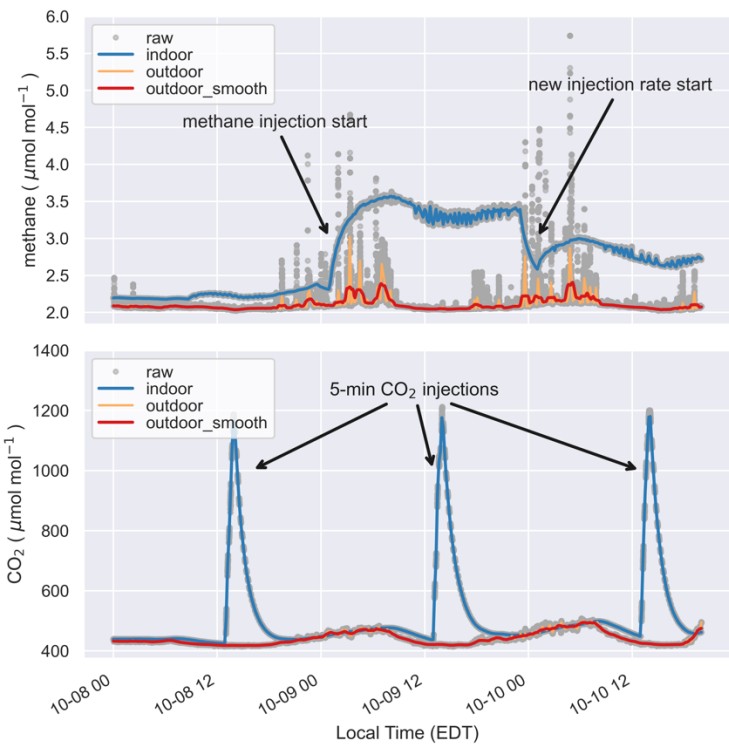


**Figure 2. Example raw and processed time series example for a three-day period with three different methane injection rates (each lasting approximately 24 h) (upper panel) and daily $CO_2$ injections (each lasting 5 min) for ACR determination (lower panel). Gray points indicate the raw (2.5-second) CRDS measurements, blue lines the indoor 5-minute means, orange lines the outdoor 5-minute median, and the red lines the smoothed outdoor data.**

**2.4 Mass Balance**

To estimate either the ACR or emission rate in the house volume, we used a mass balance approach, retaining molar units without assuming constant density across indoor and outdoor air. The governing equation for a mass balance of an inert tracer (methane, $CO_2$, or $SF_6$ in this study) in an indoor volume is (Nazaroff and Cass, 1986):



$$ER(t) = n_{air} MW \left( \frac{dX}{dt} - ACR(X_o(t) - X(t)) \right),$$ (2)

where $ER(t)$ (g h$^{-1}$) is the emission rate into the house at time $t$, $MW$ is the molar mass (g mol$^{-1}$) of the trace gas in question, and $n_{air}$ represents the moles of air inside the house volume, calculated assuming an ideal gas as $n_{air} = PV/RT$, where $P$ and $T$ are the pressure and temperature inside the house, respectively, $V$ is the volume of the house, and $R = 8.314$ J (mol·K)$^{-1}$ is
the ideal gas constant. $X(t)$ is the mole fraction (mol tracer per mol of air) within the volume, $X_o(t)$ is the mole fraction outside (both functions of time $t$), and ACR is the air change rate in house (h$^{-1}$), or $\dot{n}/n_{air}$, where $\dot{n}$ is the molar flow rate of air (mol h$^{-1}$) into or out of the volume (assumed equal).

Note that mole fraction is used in Eq. (2), rather than a mass per volume concentration, as indicated in previous work (e.g., Lebel et al. (2022)). This is necessary to account for the variation in temperatures between inside and outside during the
fall and winter season, causing the volume flow rate into and out of the house not to be equivalent. Not accounting for temperature changes during large differences between indoor and outdoor temperatures can lead to errors up to 100 % in calculating small emission rates (such as our measured quiescent methane emission rates of < 0.5 g d$^{-1}$) using this method. Similarly, as noted in Sect. 2.3, the wet mole fraction of methane or $CO_2$, rather than the dry air mole fraction, was used to account for differences in water vapor between inside and outside, which also could impact the calculation. In both cases, the
errors introduced by not using the above methods are largest when the mole fraction difference between indoors and outdoors is smallest, because small errors in the mole fraction differences have a large relative impact on the calculated emission rate. These effects are negligible when using Eq. (2) for the ACR calculation, as the indoor-outdoor mole fraction differences are very large by design during those measurements.

**2.5 Air change rate calculation**

ACRs were calculated following ASTM tracer decay method E741 (ASTM, 2023) using either $CO_2$ (ACR$_{CO2}$) or SF$_6$ (ACR$_{SF6}$) as the tracer. It was assumed the air in the house was well-mixed due to the continuous operation of the HVAC recirculation fan, based on preliminary measurements of the tracers at multiple locations in the house. After injecting the tracer such that $X \gg X_0$ and then ceasing all emissions so that $ER(t) = 0$, the decay of $X$ inside the volume is expressed as:

$$\frac{dX}{dt} = ACR(X_o - X(t)).$$ (3)

Solving this first-order differential equation with the assumption that $X_o$ is constant over the time period gives:

$$\ln(X(t) - X_o) = -(ACR)\,t + A,$$ (4)






where $A$ is the natural log of the peak mole fraction difference, a constant. For our experiments, $SF_6$ and/or $CO_2$ were injected into the house (see Sect. 2.2) for a defined period of time. This allowed mole fractions to decay with no disturbance in the house (i.e., no personnel) resulting in no indoor sources of $CO_2$. The period for the decay calculation began 10 min after the peak tracer mole fraction was measured, to avoid noisy values or sharp transitions that may occur right after the injection

ended. The end of the calculation period was determined as either the time at which $(X - X_o)$ dropped to 33 % of its peak value, or 1 h, whichever was greater. $X_o$, the outdoor mole fraction, was assumed to be zero for $SF_6$. For $CO_2$ calculations, we used a constant $X_o$ in Eq. (4) that was the average of the outdoor smoothed mole fractions over the decay time. Following Eq. (4), the ACR was determined as the slope of the regression of the natural log of the difference between the indoor time-varying mole fraction and the constant outdoor mole fraction (or zero in the case of $SF_6$) with time (Fig. 3).


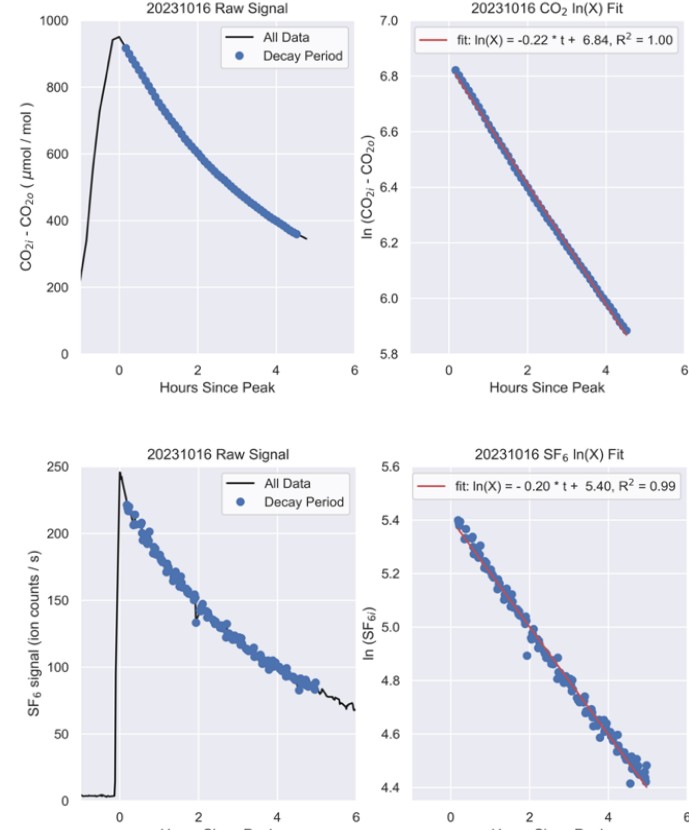

**Figure 3. Example of air change rate (ACR) calculation using the decay of $CO_2$ (top row) or $SF_6$ (bottom row) in the manufactured house. The left panels show measured $CO_2$ and $SF_6$ as a function of time with the blue points indicating the decay period for the calculation. The right panels show the linear fit to the natural log of the $CO_2$ indoor-outdoor difference and the $SF_6$ signal; uncertainty in the fit was below 1 % of the fit value. The ACR is the slope of the linear fit ($ACR_{CO2}$ = 0.22 h$^{-1}$ and $ACR_{SF6}$ = 0.20 h$^{-1}$).**




**2.6 Methane emission rate calculations**

We calculated whole-house methane emission rate ($ER(t)$ in Eq. (2)) using two different experiments. For the first experiment, methane was injected at measured rates (using a flow controller, as described in Sect. 2.2) and at different ACRs, which were controlled by either turning all mechanical ventilation off, turning on only the stove exhaust fan, or turning on the whole-house

ventilation system. The goal of this experiment was to evaluate how well the mass balance-calculated emission rate (ER$_{injection,MB}$) based on Eq. (2) could replicate the emission rate as measured by the mass flow controller (ER$_{injection,MF}$) and to quantify an error associated with the method as applied here. The second experiment consisted of a longer-term analysis of the baseline quiescent emission rate (ER$_{quiescent}$), i.e., the methane emitted from the house's gas infrastructure when there was no methane injection. The goal of this experiment was to determine how much ER$_{quiescent}$ varied over time, thus constraining how

well a single measurement of the emission rate may represent a long-term average.

Equation (2) was used to determine the methane emission rate from the house during both the methane injection (ER$_{injection,MB}$) and the quiescent experiments (ER$_{quiescent,MB}$), using the ACR values that aligned closest in time with the methane measurement. Typically, the tracer gas decay time period for the ACR calculation overlapped with the period over which the methane emission rate was calculated, with start times for the tracer decay period always within 3.5 h of the start time of the

methane emission rate calculation period. For a few days at the beginning of the experiment, the calculation was performed in the early morning (12 a.m. to 8 a.m. local time) to avoid interference from researchers in the house during the day, but was later changed to the local afternoon (1 p.m. to 7 p.m.), while also ensuring no personnel entered the house during those times. We found that conducting the experiment in the afternoon led to lower uncertainty in the outdoor mole fractions, as the outdoor mole fractions exhibited less variability during the afternoon when the atmosphere outside the house was more well-mixed

due to atmospheric boundary layer dynamics. Figure 2 illustrates an example of two early morning periods with very high outdoor methane variability while afternoon outdoor values were significantly more stable. Along with the ACR from a given day, each term in the right-hand-side of Eq. (2) was averaged over the determined time period to calculate an average methane emission rate.

One uncertainty in the emission rate estimate method is the volume to use in the calculation. Excluding the HVAC

closet and interior wall cavities the house volume is 311 m³. If these volumes are added the volume is 324 m³ (Fig. 1). The total volume measurement of the house including the exterior walls (excluding the attic and crawl space) is 340 m³. The emission rate calculation depends linearly on the volume of the space (as $n_{air}$ is the number of moles of air inside the house volume). Hence, depending on the volume chosen for the mass balance the emission rate could vary by 10 %. For this work we used a volume of 324 m³. We also note that this methodology may not capture all emissions that have made it into hidden

spaces: wall cavities, crawl spaces, attics, and other spaces not directly connected to the living volume.





# 3 Results

## 3.1 Air change rate calculation

Air change rates (ACRs) were varied by changing the mechanical ventilation in the manufactured house. Without mechanical ventilation the ACR ranged between 0.1 h$^{-1}$ and 0.5 h$^{-1}$. Turning on the exhaust fan above the gas stove increased the ACR to 245 between 0.6 h$^{-1}$ and 0.8 h$^{-1}$. Adding the whole house fan to the gas stove exhaust fan further increased the ACR to between 2.2 h$^{-1}$ and 2.5 h$^{-1}$. Comparison between the ACRs determined from the two different tracers shows most differences to be less than 20 % with a mean absolute difference of 8.1 %. Generally, larger percent differences occurred at low ACRs where small absolute differences made a large impact (Fig. 4). At the highest ACRs (over 2 h$^{-1}$) we found that using $CO_2$ as the tracer resulted in lower ACRs (4% lower on average) than using $SF_6$.


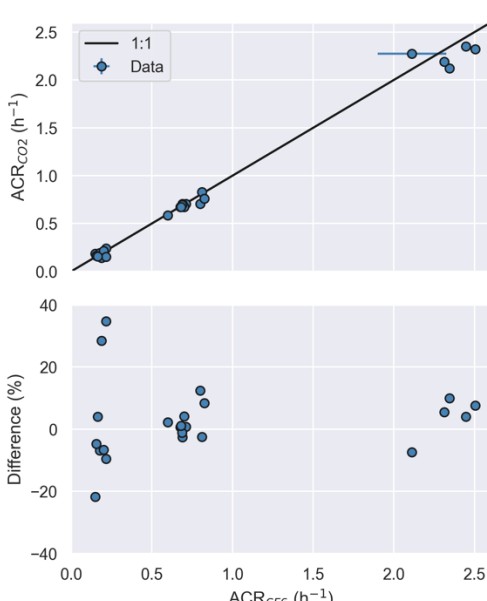

**Figure 4. Top: Air change rates calculated using $CO_2$ relative to those calculated using $SF_6$, for days when both were injected within 2 h of each other. Error bars indicate the uncertainty from the fit calculation, however most are too small to see at this scale. Bottom: Percent difference in ACR from the two methods, i.e., (ACR$_{SF6}$-ACR$_{CO2}$)/ ACR$_{SF6}$). Small ACRs showed larger relative (but smaller 255 absolute) differences.**

Nabinger et al. (2010) measured the ACR in the NIST manufactured house, finding that when no mechanical ventilation was operating the ACR was a function of both the indoor-outdoor temperature difference and wind speed. We also found that without any mechanical ventilation, the ACR correlated with the indoor-outdoor temperature difference ($R^2 = 0.44$, slope = 0.008 h$^{-1}$ °C$^{-1}$ with a slope uncertainty of 0.001 h$^{-1}$ °C$^{-1}$.) For the entire measurement period (using the daily $SF_6$- 260 calculated ACR under the conditions of no mechanical ventilation, N=48), the ACR was 0.27 h$^{-1}$ with a standard deviation of 0.09 h$^{-1}$, thus demonstrating a 32 % variability over indoor-outdoor temperature differences ranging from 0 °C to 20 °C. We



did not observe a correlation with temperature difference when either the stove exhaust fan or whole-house fan was turned on (similarly to Nabinger et al. (2010)). These results indicate that it is important to measure the ACR over the same time period as the emission rate because assuming a constant ACR could lead to significant error in the emission rate calculations.

## 3.2 Methane emission rate uncertainty estimated using injection experiments

In the first experiment, we compared methane emission rates calculated using the mass balance approach (Eq. (2), $ER_{injection,MB}$) with emission rates from mass flow-controller measurements of injected methane ($ER_{injection,MF}$). To determine $ER_{injection,MB}$, the quiescent emission rate ($ER_{quiescent,MB}$) was subtracted from the total emission calculated using the mass balance ($ER_{total,MB}$):

$$ER_{injection,MB} = ER_{total,MB} - ER_{quiescent,MB}. \tag{5}$$

Here we subtracted the mean $ER_{quiescent,MB}$ of the 35 measurements from experiments with no injection and all gas appliances turned off (0.014 g h$^{-1}$ ± 0.003 g h$^{-1}$ (1-sigma), results further described in Sect. 3.3). $ER_{injection,MB}$ generally showed 1:1 agreement with $ER_{injection,MF}$ with a slight under-estimation at low ACRs (dark blue data, Fig. 5).

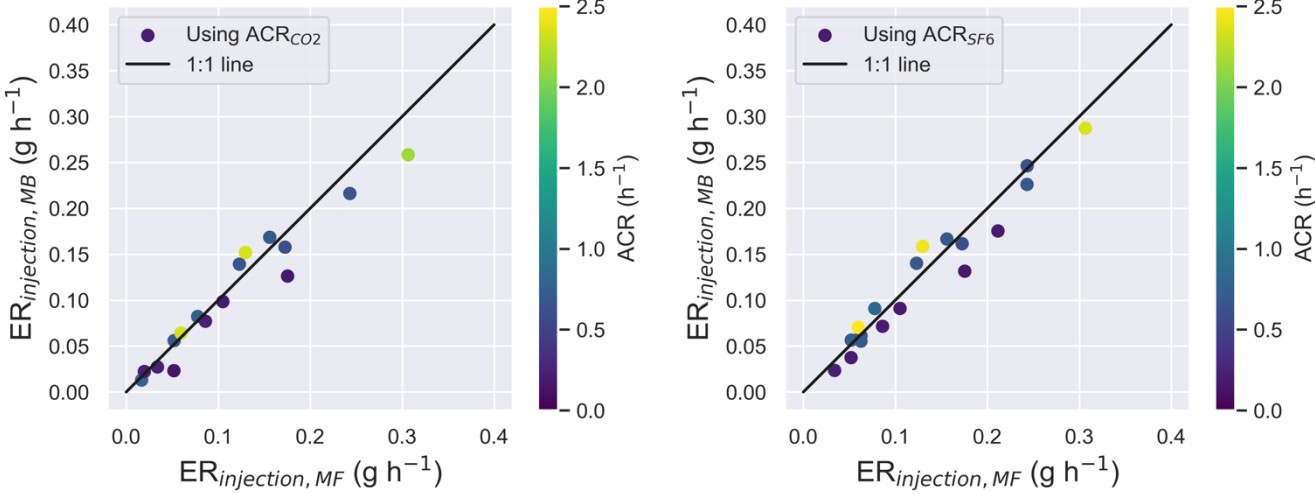

**Figure 5. Mass-balance calculated methane emission rate ($ER_{injection,MB}$) using ACRs determined from either $CO_2$ (left) or $SF_6$ (right) decay, versus the mass-flow controller measured emission rate ($ER_{injection,MF}$), colored by the ACR.**

We evaluated $ER_{injection,MB}$ from both $ACR_{CO2}$ and $ACR_{SF6}$ separately against $ER_{injection,MF}$, using bias (the mean emission rate difference ($ER_{injection,MB} - ER_{injection,MF}$)), standard deviation of offsets (SD), and root-mean-square-deviation (RMSD), as a percentage of the $ER_{injection,MB}$ (Table 1). These statistics, when evaluated separately for low and high ACRs, indicate that experiments at lower ACRs show a higher relative bias. When considering only higher ACRs, the errors are lower than at low ACRs, with RMSDs between 10 % and 14 %.




**Table 1. Performance statistics of calculated ERs for tracer injection experiments. Bias is the mean emission rate difference (ER$_{injection,MB}$ − ER$_{injection,MF}$); SD is the standard deviation of the differences; RMSD is the root-mean-square deviation of differences. All values are expressed as percentages of ER$_{injection,MB}$. Statistics are presented separately for calculations using ACRs calculated from SF$_6$ decays or CO$_2$ decays, indicated in the header.**

|  | Bias (SF$_6$) | SD (SF$_6$) | RMSD (SF$_6$) | Bias (CO$_2$) | SD (CO$_2$) | RMSD (CO$_2$) |
|---|---|---|---|---|---|---|
| **All** | -7.5 % | 18.9 % | 19.8 % | -13.2 % | 33.5 % | 35.0 % |
| **ACR ≤ 0.3 h$^{-1}$** | -28.9 % | 11.7 % | 30.8 % | -32.4 % | 47.7 % | 54.3 % |
| **ACR > 0.3 h$^{-1}$** | 3.2 % | 10.6 % | 10.6 % | -1.7 % | 14.7 % | 14.0 % |

**3.3 Quiescent whole-house emissions over time**

We determined the whole-house emission rates (ER$_{quiescent}$, with no gas appliances on and no methane injection) over a 3.5-month period. We present emission rates using the SF$_6$-derived ACR, as those were consistently available for the 3.5-month duration. The average ER$_{quiescent}$ over all 35 test days with no heating was 0.34 g d$^{-1}$ (0.014 g h$^{-1}$) with a standard deviation of 0.08 g d$^{-1}$ (0.003 g h$^{-1}$). Emission rates did not show any trend across the test period (Fig. 6). We found no correlation of

ER$_{quiescent}$ with ambient pressure or with indoor and outdoor temperature differences.

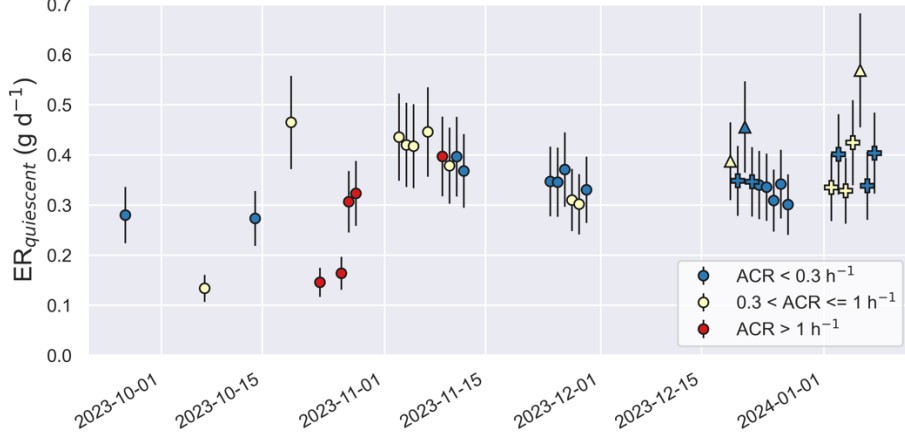

**Figure 6. Quiescent whole-house methane emission rate as measured using ACR$_{SF6}$. Error bars represent the RMSD error derived from the tracer analysis, i.e., 20 % of the calculated emission rate. Triangles represent time periods when the natural gas furnace heating was active during the averaging period; these points were excluded from the reported ER$_{quiescent}$ average. Plus symbols represent time periods where the natural gas furnace heating was active during the previous 24 hours.**

The electric house cooling system was in operation with the heating system disabled for most of the study, including for all the tracer injection experiments. For several days in late December and early January, however, the house natural gas heating system was enabled with the thermostat set to maintain a constant temperature of 20 °C (select days set at 22 °C). The

number of minutes that the heat was on was logged, and emissions during time periods with the heat active (triangles in Fig. 6) were omitted from the mean ER$_{quiescent}$ calculation. Two of the three experiments conducted with the heat on show slightly





higher emission rates than average, but one does not, and their magnitude does not correlate with the amount of time the heat was running. $ER_{quiescent}$ for days during which the heat turned on during the 24 hours prior to the experiment (plus symbols in Fig. 6) but not during the methane measurement itself do not appear to be impacted by the heating system, so they are included

in the mean. Future experiments could be designed specifically to investigate the impact of the furnace on the whole-house emission rate, including the relationship between $ER_{quiescent}$ and how long the heat was running, how much gas was consumed, and how many times the furnace cycled on or off.

## 4 Conclusions and discussion

In this study, we used the NIST manufactured house to investigate methane emission rate measurements in a residential

building. We measured air change rates (ACR) using two different tracers in a decay method, used a tracer injection method to quantify errors in whole-house methane emission rate calculations, and measured quiescent methane emissions over a 3.5-month period to examine variability. Here we discuss our results, some likely sources of error, and recommendations for future studies.

One major drawback to measuring emissions using the mass balance method described here is that there is an

assumption of emissions occurring within an enclosed envelope of known volume. Emissions occurring in crawl spaces, attics, and other areas that are not well-mixed relative to the rest of the house may not be captured, depending on varying differential pressures between the space and the sampled volume. Relatedly, the mass balance method assumes that the mixing within the enclosed space is perfect. For these experiments, we used an HVAC system fan that circulates tracers throughout the space, making this a reasonable assumption. However, that may not be the case in other houses. In addition, the actual mixed volume

must be known, as it is used directly in the emission rate calculation, so uncertainty in the volume measurement propagates directly to the results.

Several sources of uncertainty impact the measured mole fractions and their differences between indoor and outdoor air. One of the largest parameters impacting mole fraction uncertainty in our test is the variability of the outdoor methane mole fraction. Our experiment was sometimes conducted during periods of high outdoor methane variability, likely caused from

leakage in the natural gas infrastructure on the NIST campus. Similar conditions are likely to be found in dense residential and urban areas that are surrounded by methane emissions sources. Performing experiments during the daytime when these emissions were more uniformly mixed around the house reduced the variability and therefore the uncertainty in the methane mole fraction infiltrating the house. Our emission rate testing also illustrated that accounting for temperature and humidity differences between indoor and outdoor air is important when estimating emission rates from small differences between indoor

and outdoor air mole fractions. We also found that when using a high-precision CRDS instrument (1-sigma precision below 0.001 $\mu$mol mol$^{-1}$ at 0.5 Hz), analyzer precision is not a significant factor in the error in the emission rate. However, if using lower-precision instruments, instrument precision and possible drift over experimental time scales should be considered in the analysis. The uncertainties described above are most significant when methane enhancements inside the house are small; larger



methane enhancements inside (due to low ACR and/or high emission rates) would reduce the impact of uncertainties on the
mole fractions.

We found that using $CO_2$ as a tracer for the ACR calculation was nearly equivalent to using $SF_6$, with ACRs agreeing within 8 % on average. During ACR tracer decay measurements, care should be taken to eliminate or at least minimize any additional ventilation of the house, such as personnel entering or exiting the spaces, or opening any doors or windows. Additional caution must be exercised when using $CO_2$ to eliminate interference from other $CO_2$ sources (such as occupants or
combustion in the house) and to inject enough $CO_2$ to achieve a large signal relative to outside variability. Over the course of the 3.5-month study, the ACR in the manufactured house with no mechanical exhaust was variable (32 %) under the conditions of no mechanical ventilation. Therefore, timing the ACR measurement, especially for buildings without mechanical ventilation, should be as close as possible and during similar conditions as the emission rate measurement to help prevent variability in ACR from introducing error in the calculated emission.

Unlike the ACR, $ER_{quiescent}$ could not be measured simultaneously with the injected methane, as the mass balance method necessarily measures the total methane emission into the house. Thus, $ER_{quiescent}$ had to be subtracted from the total measured emission rate which included both the methane injection and the quiescent house emissions. We subtracted the average $ER_{quiescent}$ measured during the entire experiment from the total emission rate measured in each methane injection experiment, so day-to-day variability in $ER_{quiescent}$ contributed to error in the calculation.

Our measurements of whole house quiescent methane emission rates in the manufactured house averaged 0.34 g $d^{-1}$, similar to the mean of 0.5 g $d^{-1}$ found by Fischer et al. (2018) in a study of 75 California single family homes. Variability in this study from day to day was 0.08 g $d^{-1}$ (standard deviation), or 24 % of the quiescent emission rate, similar to the 20 % estimated uncertainty from the $CO_2$ and $SF_6$ experiments, indicating that this temporal variability is not distinguishable from our expected methodological error.

Our methane tracer injection results showed RMSD of 20 % on calculated emission rates, averaged over a range of ACRs and injected emissions. Errors were higher when ACRs were between 0.2 $h^{-1}$ and 0.3 $h^{-1}$ and not correlated to the magnitude of the indoor-outdoor methane differences. Specifically, emission rates calculated under the low ACR conditions (0.2 $h^{-1}$ to 0.3 $h^{-1}$) were generally biased low over a range of emission rates. It is possible that the ACR calculated using the tracer decay method exhibited a low bias at lower ACRs, warranting further investigation. In general, while typical U.S. single-family
homes exhibit ACRs between 0.2 $h^{-1}$ and 1 $h^{-1}$ (Nazaroff, 2021), newer homes are often built with reduced ACRs to achieve space heating and cooling efficiencies and reduce energy use. Given our results, when making measurements in spaces with low ACRs, it may be useful to turn on some mechanical ventilation for the test period. Higher ACRs also allow the methane mole fraction to come to equilibrium faster, which may make the testing quicker, but care should be taken if overall indoor enhancements are small, because other errors have larger relative impact when enhancements are small (as discussed above),
and instrument detection limits may be approached. If possible, conducting methane injections tests prior to measuring emission rates from different houses would give the best estimate of uncertainty for each test given the different conditions likely to be encountered in a real-world experiment.



        Additional measurements of whole-house emission rates of methane including from houses with different characteristics (i.e., age, location, and size) would be helpful to supplement the research performed here and to better inform national

inventories (Karion et al., 2024). Future work could analyze the effects of gas appliance usage such as gas furnaces or stoves on whole-house emission rates. The average emission rates would likely increase with the use of these appliances (Lebel et al., 2022; Lebel et al., 2020), hence investigating the relationship would be informative. Relationships relative to gas usage especially could shed light on recent top-down methane studies showing a relationship between emissions and monthly natural gas use over an entire urban area (Sargent et al., 2021; Zeng et al., 2023; Karion et al., 2023).

*Code and data availability.* The code and data used in this study are available by request to Anna Karion (anna.karion@nist.gov).

*Author contributions.* AK, DP, and MFL designed the experiments and MFL, RR, AK, and TB carried them out. AK prepared the manuscript with contributions from all co-authors.


*Competing interests.* The authors declare that they have no conflict of interest.

*Acknowledgements and support.* This work was supported by the NIST Greenhouse Gas Measurements Program and the NIST Engineering Laboratory.

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
