# Peer review of "Methodology and uncertainty estimation for measurements of methane leakage in a manufactured house"

_EGUsphere, 2024_

## Referee Comment (RC1)

I read manuscript titled: **Methodology and uncertainty estimation for measurements of methane leakage in a manufactured house,** with great interest. First of all, I have to send my appreciation to the authors for their thinkings related to the details during the measurements and evaluation. The authors performed measurements in a manufactured house using control release experiment of two tracers quantified quiescent emissions from the house during a 3.5-month study. I recommend this manuscript to be published after addressing the following comments.

**General comments**:

- When comparing with other studies and inventories please make sure that you compare gas leaks to gas leaks and / or unburnt emissions to unburnt emissions.
- More details are required to clarify the methods and instruments used.
- It would be great also to have statistics about US residence living in a single household and in a complex.
- Possibly further statistics can be added to the manuscript understanding impact of e.g. wind speed on the Air Change Rates (ACRs) and error related to emission quantifications.

**Detailed comment**:

L27 – As methane reduction is meant to slow down global warming at shorter period of time, it is recommended to write global warming potential of more than 80 times relative to $CO_2$ over 20-year time horizon.

L43 – This long-tail has been observed in almost all studies resulting that few emitting sources contribute significantly to the total emissions. I am wondering how many houses do you need to measure to possibly see this long-tail and how this differs from a state to another? What sampling strategy do you think can address this concern?

L43 – In the calculation of the 15% contribution to total methane emission in California, unburnt methane from pilot is included. According to the Fischer et al. (2018) emissions from pilots contribute 30% to the household emissions. As it is written here, it reads that the quiescent emissions contribute 15% to total, please clarify. On another note, I assume pilots were off in your studies, is that right?

L74 – It is great that the study was conducted over relatively long period, however I am wondering how quiescent emissions can show seasonal variability. Maybe temperature could be a reason but then my question is: is it possible that leaks are emitting higher in warmer season due to possible thermal expansion?

L87 - As I search over the net and compared ACRs, it seems that manufactured house is among house which are air-tight. Is it possible to apply your method to houses with bigger ACR? Possibly those houses are among bigger quiescent emitters due to older/aged technology within the natural gas piping?

L90:92 – Please provide details about manufacturers of the instruments you used in your study.

L126 – Please add details about the flowmeter, see the comment above.

L135 – Please add 'to' after 'in order'.

L185 – Please see the comment for Figure 2.

L240 – If possible, could you please elaborate how it would be possible to capture those possible emissions? I am wondering how significant those emissions are relative to total emissions.

L257 – I don't see an analysis over impact of wind speed and ACR in your manuscript. Would it be possible to see how wind speed impact the ACR and/or favourable wind speed in which it is recommended to perform the emission measurements? Maybe you can look at the error during the injection period and check the results against wind speed logs. It's not clear to me where you possibly have access to the wind speed logs, the stations in Arlington, Virginia (Karion et al., 2020) is abit far though for this analysis.

L320 – This is right, please elaborate, see comment related to L240.

L361 – This ACR falls into the range for the tight-air houses, how representative this range can be for the normal houses in use? On another note, can you use this method for the high rise buildings If possible, please elaborate in the manuscript accordingly.

L365 – Please see Figure 7 from this link for these link: https://www.ncbi.nlm.nih.gov/pmc/articles/PMC5369024/

**Figures**:

Figure 1 – It seems like that the injection and sampling points are relatively close to each other compared to the size of the house. How did you ensure that the tracers are well-mixed during in the volume of the house during sampling? Could you please add information about strength of the fan used for the indoor circulation?

Figure 2 – Just out of curiosity, what are those spikes in the outdoor raw measurements happening around midnight on both days?

Figure 6 – (similar to the comment for L310:313) the triangle dots show that while natural gas furnaces are active, total emissions from the house is higher, this can be investigated further to understand methane emissions from active furnaces including incomplete combustion.

Note 1: Would it be possible to leave the manufactured house in one state over long time and see if the house's gas meter change over time indicating amount of quiescent emissions over a long time to drive daily emissions? I assume the quiescent emissions per day is small enough not to be observed by a gas meter.

Note 2: How often the gas-related pipelines and utilities in this manufactured house go under maintenance? Is it similar to the maintenance frequency of normal houses?

Note 3: Further research efforts are required to complete the indoor household emissions to achieve representative sample size for different types of houses including different natural gas-related infrastructure.

**Reference**:

Fischer, M. L., Chan, W. R., Delp, W., Jeong, S., Rapp, V., and Zhu, Z.: An Estimate of Natural Gas Methane Emissions from California Homes, Environmental Science & Technology, 52, 10205-10213, https://doi.org/10.1021/acs.est.8b03217, 2018.

Karion, A., Callahan, W., Stock, M., Prinzivalli, S., Verhulst, K. R., Kim, J., Salameh, P. K., Lopez-Coto, I., and Whetstone, J.: Greenhouse gas observations from the Northeast Corridor tower network, Earth Syst. Sci. Data, 12, 699-717, https://doi.org/10.5194/essd-12-699-2020, 2020

---

## Author Comment (AC1)

Response to Reviewer 1 Comments (RC1):

Reviewer comments are in italics, our responses are in plain text. Changes to the main text are included in red.

*I read manuscript titled:* **Methodology and uncertainty estimation for measurements of methane leakage in a manufactured house***, with great interest. First of all, I have to send my appreciation to the authors for their thinkings related to the details during the measurements and evaluation. The authors performed measurements in a manufactured house using control release experiment of two tracers quantified quiescent emissions from the house during a 3.5-month study. I recommend this manuscript to be published after addressing the following comments.*

We thank the reviewer for the detailed reading and review of our manuscript.

***General comments:***
*- When comparing with other studies and inventories please make sure that you compare gas leaks to gas leaks and / or unburnt emissions to unburnt emissions.*

We have now clarified the definition of "quiescent" emissions in the abstract to explain that it includes emissions from pilot lights but excludes inefficient combustion from appliances are in use. This is the same definition Fischer et al. used for "quiescent whole house emissions", which is the value we compare with ours, although our study does not include pilot lights (the appliances do not have them, and we now state this in the manuscript Secdtion 2). They found that ~30% of those quiescent emissions were from pilot lights.

*- More details are required to clarify the methods and instruments used.*

We have addressed this in the specific comments below.

*- It would be great also to have statistics about US residence living in a single household and in a complex.*

The American Housing Survey of the US Census indicates that 61 % of US housing structures are 1-unit detached housing, and 5.5% are manufactured homes (mobile homes), such as the one studied here.  The remainder are multi-family and attached housing, such as townhomes or complexes. This is now included in the introduction with some context as to how we expect our results to be extensible to such single-family detached homes:

"We note that while only 5.5 % of US housing units are manufactured houses (i.e., mobile homes) such as this one, over 61 % are 1-unit detached homes (US Census American Community Survey, 2023); the methodology described here is relevant to detached houses in which mixing can be ensured or imposed through mechanical systems."

*- Possibly further statistics can be added to the manuscript understanding impact of e.g. wind speed on the Air Change Rates (ACRs) and error related to emission quantifications.*

We have addressed these in the specific comments below.

***Detailed comment:***

*L27 – As methane reduction is meant to slow down global warming at shorter period of time, it is recommended to write global warming potential of more than 80 times relative to CO2 over 20-year time horizon.*

This has been added.

*L43 – This long-tail has been observed in almost all studies resulting that few emitting sources contribute significantly to the total emissions. I am wondering how many houses do you need to measure to possibly see this long-tail and how this differs from a state to another? What sampling strategy do you think can address this concern?*

This is a good point, but one which is outside the scope of our work given that we only measured emissions from a single house and cannot speculate as to the nature of the long-tailed distribution. We do have a statement in the conclusion regarding future work that "Additional measurements of whole-house emission rates of methane including from houses with different characteristics (i.e., age, location, and size) would be helpful to supplement the research performed here and to better inform national inventories."

*L43 – In the calculation of the 15% contribution to total methane emission in California, unburnt methane from pilot is included. According to the Fischer et al. (2018) emissions from pilots contribute 30% to the household emissions. As it is written here, it reads that the quiescent emissions contribute 15% to total, please clarify. On another note, I assume pilots were off in your studies, is that right?*

Yes, pilot lights were off in our studies (more accurately, none of the gas appliances have pilot lights). We have added this in a sentence in the Methods section. As we interpret the results in Fischer et al., the 15% contribution stated there includes all sources: passive leaks, pilot lights, and steady-state appliance use. Pilot lights were ~30% of the quiescent whole house totals. We have now further clarified the 15% statistics in the introduction:

"Measurements of emissions from 75 homes in California indicated that typical homes have quiescent methane emission rates (including emissions from pilot lights) of <1 g d$^{-1}$, but with some as high as 10 g d$^{-1}$, with total emissions from residential natural gas, i.e., including both quiescent emissions and emissions from steady operation of appliances, accounting for approximately 15 % of California's natural gas emissions (Fischer et al., 2018)."

*L74 – It is great that the study was conducted over relatively long period, however I am wondering how quiescent emissions can show seasonal variability. Maybe temperature could be a reason but then my question is: is it possible that leaks are emitting higher in warmer season due to possible thermal expansion?*

We did not find any correlation of the emissions with either indoor or outdoor temperature. Now this is stated clearly in 3.3 along with the range of outdoor temperatures, "We found no correlation of $ER_{quiescent}$ with outdoor temperature (which ranged from 0 °C and 26 °C during the experiments), indoor temperature, or ambient pressure. "

*L87 - As I search over the net and compared ACRs, it seems that manufactured house is among house which are air-tight. Is it possible to apply your method to houses with bigger ACR? Possibly those houses are among bigger quiescent emitters due to older/aged technology within the natural gas piping?*

It is true, this manufactured house has a low ACR as it was used in weatherization studies (https://tsapps.nist.gov/publication/get_pdf.cfm?pub_id=906322). However, we do believe that our method applies to houses with higher ACR, because we performed our tracer experiments at the very low default ACR but also at intermediate and high ACRs induced by mechanical ventilation.  We also separate out our errors by ACR in Table 1 because we found lower errors at high ACRs. The method for measurement should be robust across ACRs.

*L90:92 – Please provide details about manufacturers of the instruments you used in your study.*
*L126 – Please add details about the flowmeter, see the comment above.*

NIST's policy is to not name specific products to prevent promoting of specific brands or technologies.  As a government agency we do not want to be perceived as directly influencing the market, and in this case we do not want to tie the measurement method we are describing to a specific manufacturer's product. We can individually let the reviewer know this information if they think it is necessary to review the paper.

*L135 – Please add 'to' after 'in order'.*

This sentence has now been removed because we have changed the method to use the mean of the outdoor concentrations based on a comment from Reviewer 2.

*L185 – Please see the comment for Figure 2.*

See the response under the comment for Figure 1 (I believe this was the question, regarding how we assumed the tracers injected into the house were well-mixed).

*L240 – If possible, could you please elaborate how it would be possible to capture those possible emissions? I am wondering how significant those emissions are relative to total emissions.*

The way to capture emissions within wall cavities, crawl spaces, etc., would be to incorporate their volumes into the mixed volume of the main house space, i.e. opening them up to the rest of the house and ensuring mixing through those spaces into the main house. Alternatively, one could negatively pressurize the house (e.g., using a blower door) so that emissions from those spaces enter the well-mixed volume that is being measured. We have now mentioned this in this part of the manuscript, the end of Section 2.6: "To capture the emissions in such spaces using this methodology, the volume of the house would either need to be negatively pressurized relative to the hidden spaces or the hidden spaces would need to be connected (opened) to the main volume of the house with adequate mixing through the entire volume to ensure uniform concentrations. "

*L257 – I don't see an analysis over impact of wind speed and ACR in your manuscript. Would it be possible to see how wind speed impact the ACR and/or favourable wind speed in which it is recommended to perform the emission measurements? Maybe you can look at the error during the injection period and check the results against wind speed logs. It's not clear to me where you possibly have access to the wind speed logs, the stations in Arlington, Virginia (Karion et al., 2020) is abit far though for this analysis.*

We thank the reviewer for this comment, this was a very good idea.  We have now obtained and analyzed weather station data from NOAA's ASOS network from the airport 6.8 km away, which includes both outdoor pressure and wind speed. We use this pressure now instead of the pressure from Arlington, VA, as it is closer.  The wind speed was indeed positively correlated with ACR, i.e. higher ventilation rates at higher wind speed as would be expected from previous literature published on this house (https://tsapps.nist.gov/publication/get_pdf.cfm?pub_id=906322), and this is included now in Section 3.1.  We also looked at whether errors were higher during periods of high wind variability, by looking at correlations between relative errors (and absolute errors) with wind speed and the wind speed standard deviation over the measurement period, and found no correlation.

*L320 – This is right, please elaborate, see comment related to L240.*

See response above related to L240.

*L361 – This ACR falls into the range for the tight-air houses, how representative this range can be for the normal houses in use? On another note, can you use this method for the high rise buildings If possible, please elaborate in the manuscript accordingly.*

Indeed, the ACR for this home is on the low end of typical US homes (Nazaroff, 2021), which range from 0.2 $h^{-1}$ to 1 $h^{-1}$, as mentioned in the Discussion, but it is not atypical of newer homes that are built specifically to have low ACRs to increase energy efficiency.

On the question of high-rise buildings, this method would not work as outlined here. Because the manufactured house measured here is well-mixed we can treat it as a single zone (for this specific tracer test) where the only CH4/CO2 sources are either inside or outside. This method would not work in a high-rise because of possible heterogeneous distribution of emission sources in the several zones where air could be sourced from when measuring CH4/CO2 in a single apartment. We have now added this discussion to the conclusion section:

"We also note that the requirement of a well-mixed volume prohibits the use of this method on high-rise buildings, such as multi-family complexes. Interzonal mixing (i.e., mixing between apartments or hallways) may make major contributions to the chemical composition of indoor air in any given space (e.g., an apartment) within a high-rise. This method would not work over the entire multi-unit high-rise because of possible heterogeneous distribution of emission sources and non-uniform concentrations across different zones. However, if individual well-mixed zones can be pressurized relative to other units, this approach may still yield individual zone (not whole building) emission rates."

*L365 – Please see Figure 7 from this link for these link:*
*https://www.ncbi.nlm.nih.gov/pmc/articles/PMC5369024/*

We have added this reference to our paper discussion of the range of ACRs.

***Figures:***

*Figure 1 – It seems like that the injection and sampling points are relatively close to each other compared to the size of the house. How did you ensure that the tracers are well-mixed during in the volume of the house during sampling? Could you please add information about strength of the fan used for the indoor circulation?*

CO2 and SF6 were injected into the central HVAC system and thus dispersed throughout the house through the air vents. The HVAC system fan was always on, even when cooling was not active. Previous research in this facility has shown that this method mixes the tracer throughout the house within a few minutes (citation?). Methane was directly injected into the house at a fixed location near the HVAC system but not into the intake (the figure and caption have now been edited to explain this). However, even though the methane was not dispersed through the HVAC, the HVAC system did serve to generally mix air throughout the house. Because our methane measurements were made after the methane concentration reached steady-state (hours after initial injection), we assume that it is well-mixed in the house by this point. We have now expanded this explanation in the Methods section and added the fan speed of 2000 m^3/hr in the text.

"Both CO2 and SF6 were injected into the HVAC air intake so that they were dispersed though the vents throughout the house; previous research in this house has shown that the tracers injected this way disperse (sufficiently for the ACR calculation) within 10 minutes."
And
"During the methane tracer experiments, methane was injected at a constant rate for a 24-h period at a location near the HVAC system (Figure 1) but not into HVAC intake. Methane measurements for the experiments were made well after injection began (at least 3 h) and after near-steady-state had been achieved in the concentrations; we assume it was sufficiently well-mixed in the volume by the HVAC fan system by that time."

*Figure 2 – Just out of curiosity, what are those spikes in the outdoor raw measurements happening around midnight on both days?*

We haven't traced the source of these spikes, but we believe they are leaks from the natural gas lines that are supplying a Fire Research Facility near the house. We do not state this in the paper as this is still speculation, but we do intend to find and report this.

*Figure 6 – (similar to the comment for L310:313) the triangle dots show that while natural gas furnaces are active, total emissions from the house is higher, this can be investigated further to understand methane emissions from active furnaces including incomplete combustion.*

This is a line of research planned for the future.  We were not able to further investigate this before spring arrived in this study. One issue is that the house itself contained many analyzers and equipment and the weather was not that cold, so that it was quite warm indoors even without the furnace running.  Running the furnace would have meant overheating the house and equipment. We intend to repeat the experiment with only the methane & CO2 analyzer in the house and during colder weather, and add sampling in the flue of the furnace.  We mention this here (and as the reviewer noted, previously this was line 310), "Future experiments could be designed specifically to investigate the impact of the furnace on the whole-house emission rate, including the relationship between $ER_{quiescent}$ and how long the heat was running, how much gas was consumed, and how many times the furnace cycled on or off."

*Note 1: Would it be possible to leave the manufactured house in one state over long time and see if the house's gas meter change over time indicating amount of quiescent emissions over a long time to drive daily emissions? I assume the quiescent emissions per day is small enough not to be observed by a gas meter.*

This is a good idea, but we did not investigate this, assuming that indeed the emissions would be too small over this time frame.

*Note 2: How often the gas-related pipelines and utilities in this manufactured house go under maintenance? Is it similar to the maintenance frequency of normal houses?*

The maintenance in the house on the gas-related pipelines or other systems if typical of a normal house, in that repairs are made when components fail.

*Note 3: Further research efforts are required to complete the indoor household emissions to achieve representative sample size for different types of houses including different natural gas-related infrastructure.*

Indeed, we recognize this and see this study as purely a method description and uncertainty analysis for a single measurement. We note in the conclusion that "Additional measurements of whole-house emission rates of methane including from houses with different characteristics (i.e., age, location, and size) would be helpful to supplement the research performed here and to better inform national inventories."

*Reference:*
*Fischer, M. L., Chan, W. R., Delp, W., Jeong, S., Rapp, V., and Zhu, Z.: An Estimate of Natural Gas Methane Emissions from California Homes, Environmental Science & Technology, 52, 10205-10213, https://doi.org/10.1021/acs.est.8b03217, 2018.*
*Karion, A., Callahan, W., Stock, M., Prinzivalli, S., Verhulst, K. R., Kim, J., Salameh, P. K., Lopez-Coto, I., and Whetstone, J.: Greenhouse gas observations from the Northeast Corridor tower network, Earth Syst. Sci. Data, 12, 699-717, https://doi.org/10.5194/essd-12-699-2020,*

---

## Author Comment (AC2)

Response to Reviewer 2 Comments (RC2):

Reviewer comments are in italics, our responses are in plain text. Changes to the main text are included in red.

*Top-down (atmospheric) and bottom-up (inventory) methods of determining greenhouse gas emissions are complementary approaches, with the goal of providing feedback as to the most effective mitigation strategies. There is a significant discrepancy between these approaches for methane, and in-home emissions are probably a major contributor. The EPA methane inventory now includes an estimate of these emissions, but the emission factors are all based on one study. This manuscripts documents controlled release experiments to optimize the methods to determine these emission factors and to quantify the uncertainty using the mass balance method which quantifies whole-house emissions rates as a sum of all of the leaks in the house. This work is an important first step, laying the groundwork for future studies of different types of homes in different cities (with different types and ages of infrastructure) and in different climates. The manuscript is well-organized and well-written and I feel it should be published with very minor changes.*

We thank the reviewer for their detailed reading and review of the manuscript.

*Abstract: Don't need to specify "recently" AND "latest".*

Yes --"recently" has been removed

*Section 2: Confusing wording to discuss the overall 3-part plan in the previous paragraph and then specify measuring whole-house emissions rates of methane (part 2 of the plan) as the first sentence of the methods. Suggest changing " We measured whole-house emission rates of methane in a manufactured ..." to a more general statement along the lines of "We performed the controlled experiments described above in a manufactured house..."*

This change has been made as suggested.

*Section 2.1: "A flush pump pulled air at 10 L min-1 through whichever inlet line the valve was set to." Awkward wording. Maybe "through the appropriate inlet line"*

This change has been made as suggested.

*I'm confused about the flowrates but I think it's just that there are three total tubes: indoor for CO2/CH4, outdoor for CO2/CH4, and indoor for SF6. First two lines: 10 L/min flush, CRDS sipping. Third line: PTR-MS flowrate of 15 L/min with inlet flowrate of 120 mL/min. Should this be PTR-MS flush rate of 15 L/min with inlet flowrate of 120 mL/min?*

This was poorly worded. The line has been edited regarding the PTR-MS flow as follows: "The PTR-MS sampled $SF_6$ through 8 mm (inner diameter) PFA tubing with an inlet flowrate of 120 mL min$^{-1}$ from a line that was flushed at 15 L min$^{-1}$."

*The gas heating component of the furnace and the HVAC gas heating element are the same things, so the sentences in lines 113-115 in this section were confusing to me. I think the tracer experiments were in October(?) and then the heating element was turned on in December during quiescent emissions measurements.*

Exactly. This has now been re-worded to be clearer. "While the HVAC fan was on for all experiments, the gas heating furnace was turned off for the tracer experiments, which ended in October. The gas heating was turned on in late December,..."

*How far away is the weather station in Arlington, VA?*

The weather station in Arlington, VA is about 27 km away. We have now obtained weather data from the local airport only 6.8 km away from NOAA ASOS, and are using that for pressure as well as for wind speed in this new draft. Noted now as "the outdoor pressure from a weather station at the Montgomery County Airpark 6.8 km away (NOAA, 2024) was used for both indoor and outdoor molar calculations."

*It's obvious, but consider specifying that the windows were closed.*

Yes- now added to the methods: "All tests were performed with all windows and doors closed".

*Section 2.3: You used mean indoor, but median outdoor. I'm not sure which is correct here. The spikes are real, so isn't the mean more appropriate since some of that air exchanges with the indoor air? During the afternoon, it's hopefully a small difference. How different would the results be if you used mean for outdoor?*

We thank the reviewer for this comment. We originally used the median in the assumption that the outdoor spikes were localized near the outdoor inlet and would not have infiltrated into the house. However, we had no evidence of that and now believe it is just as likely that the spikes were caused by elevated methane around the entire house that likely entered the space. We have revised our calculations using the mean of the 5-minute outdoor samples, and find a (very) slight improvement in our statistics, with the overall RMSE dropping to 19% from 20%. All figures and tables have been re-generated, and the text revised accordingly.

*Section 2.6: For the first experiment, does the ERquiescent have to be taken into account? I see that you do mention that in Section 3.2 (This should be moved to the methods.)*

We have now added a sentence in 2.6: "We note that the average $ER_{quiescent}$ from the second experiment was subtracted from the total ER in the first experiment to determine $ER_{injection,MB}$."

*Section 3.1 You didn't find any correlation with wind speed like Nabinger et al did?*

We have now obtained wind data from a local airport and indeed we do find a correlation between wind speed and ACR now included here: "We also found that without any mechanical ventilation, the ACR showed a positive correlation with both the indoor-outdoor temperature difference ($R^2$ = 0.44) and wind speed at the nearby airport ($R^2$ = 0.41)."

*Fig 6: maybe should expect error bars to be larger on the blue symbols (low ACR)*

This was a suggestion we considered but ultimately we have decided to retain the total average RMSE (19%) for these error bars. While we calculated the errors (RMSE in Table 1) for the different ranges of ACR, we did not evaluate all possible ACRs because we could not generate ACRs smoothly across the covered range. Therefore, we hesitate to assign a different uncertainty for different ACRs arbitrarily because we do not really know the beginning or end of the range where the ACR is less certain. We hope to be able to better ascertain the relationship between ACR and error in the future.

*Fig 6: The average seems lower in October than the rest of the analysis period.*

This seems to be the case, but we do not really have enough statistics to analyze a possible reason for this, as the quiescent ER in October was only measured a few times.

*Line 302 in Section 3.3 Consider rewording "The electric house cooling system was in operation with the heating system disabled for most of the study, including for all the tracer injection experiments."to " The electric house HVAC system was in operation for all of the study, with the heating system disabled for most of the study, including for all the tracer injection experiments. " or something along those lines.*

This change has been made.